# Evaluation of the operation status of the residential land market in Beijing-Tianjin-Hebei region of China and its spatiotemporal pattern characteristics

Xin Chen[1,2], Can Li[1,2]*, Huixia Li[3], Jingfeng Ge[1,2]*, Wengang Wang[1,2], Pengfei An[1,2]

**1** School of Geographical Sciences, Hebei Normal University, Shijiazhuang, Hebei Province, China, **2** Hebei Key Laboratory of Environmental Change and Ecological Construction, Shijiazhuang, Hebei Province, China, **3** Hebei Academy of Social Sciences, Shijiazhuang, Hebei Province, China

* lican3008@126.com (CL); gejingfeng@hebtu.edu.cn (JG)

**Data Availability Statement:** All relevant data are within the paper.

**Funding:** National Natural Science Foundation of China (project No.41471090) and Science

## Abstract

Exploring the operation status and patterns of urban land markets is an important theoretical and practical topic for promoting coordinated socio-economic development. In this study, the operation status of the residential land market in the Beijing-Tianjin-Hebei region and the characteristics of its pattern were analyzed using the composite index method and the 3σ rule of the normal distribution and taking the 174 counties in Beijing, Tianjin, and Hebei, China, as the research objects. The results show that ① Beijing, Tianjin, Langfang, Zhangjiakou, and Baoding residential land market state composite indexes are all in the middle to upper levels in the Beijing-Tianjin-Hebei city cluster, while Qinhuangdao, Handan, and Chengde residential land market state composite indexes are generally low. The harmony between the residential land price and national economy, the market supply and demand balance, and the structural balance may become the main factors affecting the healthy development of the residential land market in Beijing and Tianjin. ② The proportion of counties with "healthy" residential land market in all dimensions and overall market status reached over 64%, and the residential land market in the Beijing-Tianjin-Hebei region is running well. The rapid increase in residential land prices from 2016 to 2020 was an important driver of the increased heat in the residential land market across the region. ③ The residential land market in the counties around Beijing and Tianjin is significantly hotter than in other regions, and there is an obvious polarization effect in the operation state of the residential land market in the Beijing-Tianjin region. The residential land market is generally cold in the counties in the southern and northeastern parts of the region and other peripheral areas, and there is a risk of marginalization in the development of the residential land market in the counties in the peripheral areas. ④ Both the hot and cold residential land market states exhibit spatial clustering characteristics. Most of the clusters are not consistent with the municipal administrative boundaries, and the states of the residential land market in neighboring counties are very similar.

Foundation of Hebei Normal University (project No. L2021B29). Jingfeng Ge is the owner of National Natural Science Foundation of China (Fund No.41471090) and Can Li is the owner of Science Foundation of Hebei Normal University Foundation (project No.L2021B29). The funders played a relevant role in this article, Jingfeng Ge has done Writing-review & editing; Can Li has done project management and supervision.

**Competing interests:** The authors have declared that no competing interests exist.

# Introduction

Land resources are the carriers of social and economic activities, and the degree of their rational allocation is directly related to the urbanization process, sustainable economic development, and the improvement of people's livelihood [1]. As one of the countries with the fastest urban land expansion [2, 3], China's real estate industry has become a pillar industry for economic growth and an important source of local fiscal revenue in all regions, but at the same time, the problems of land resource management and use are becoming increasingly prominent. For example, the expansion of urban construction is accompanied by the disorderly spread and waste of land, which has led to inefficient land use, soaring land prices, and environmental degradation [4–6]. As the source of the real estate market, the healthy operation of the residential land market is crucial to the sustainable development of the real estate industry, and the healthy development of residential land prices is also directly related to the coordinated development of society, the enhancement of residents' happiness, and the construction of a harmonious society [7]. Therefore, the operation mechanism of the residential land market has become a hot area of concern for the government and academia in recent years. The Beijing-Tianjin-Hebei urban agglomeration is one of the regions with the most dynamic economy, the highest degree of openness, the strongest innovation capacity, and the largest population absorption in China, and it is also an important engine pulling the economic development of China. The serious divergence of land market status is a prominent manifestation of the unbalanced development of Beijing-Tianjin-Hebei region, which has become a difficult problem in the current coordinated development of Beijing-Tianjin-Hebei region. Due to the different geographical locations, urban planning, social culture and economic development levels, different cities may also have obvious differences in the rhythm of land supply and demand, land supply structure, land supply efficiency and the degree of land marketization in the process of urban construction. In February 2014, the Chinese government officially proposed the strategy of coordinated development of Tianjin and Hebei [8] to analyze and evaluate the operation status and laws of the residential land market in each city and formulate inter-city The key issue that needs to be solved for the coordinated development of Beijing-Tianjin-Hebei is to analyze and evaluate the operation status and laws of residential land market in each city, and to formulate the policy of differential residential land market regulation between cities.

Regarding land market evaluation studies, scholars have focused on different types of land demand, locational competition, and balanced land use structure in the early days [9]. In recent years, studies related to land market evaluation have tended to diversify, focusing on two main aspects.

One is the review of land market operation mechanism and related policies. The studies are mostly based on the concept and attributes of land in economics, and review the connotation [10–12] and operation mechanism [13–15] of land market. Second, the evaluation of land market operation efficiency, most scholars consider the degree of land marketization [16], the effectiveness of macro-control policies [17], and the size of its effect on socio-economic development [18–20] as important factors in evaluating land market operation efficiency. The primary market is defined as the urban land market in which the state is directly involved in the operation [21]. Regarding the evaluation of the operation status of the primary land market, some studies focus on single-factor evaluation [22–26], based on socio-economic indicators, including the effective land supply rate, the scale of land supply and the structure of land supply, to achieve the measurement of the level of land marketization. At present, more scholars tend to measure the degree of land market development based on a multi-indicator system in order to reflect the market development more objectively [27–30]. In addition, some scholars

have also carried out the evaluation of land market development from other perspectives [31–33], using various methods such as 3σ to distinguish the operation status of land markets in different cities. In terms of research methods, scholars have tried to improve and innovate the evaluation methods, such as the topologizable meta-evaluation method and econometric model [34]. The determination of indicator weights is mostly based on factor analysis method, coefficient of variation method, entropy weight method and other measurement methods, and the evaluation of land market state is mostly based on multi-factor comprehensive evaluation method.

To sum up, the current land market evaluation system tends to be diversified and does not form a unified standard, and there are relatively few comprehensive multi-factor evaluation systems that can comprehensively characterize the supply and demand mechanism, price mechanism and competition mechanism. Most studies in the past have evaluated the health of the land market from the perspective of complementary equilibrium between the real estate market and economic, social, and natural [35] factors, and its health connotation includes various aspects such as the size of the market, the strengths and weaknesses of the market structure, the speed of development, and the high and low land prices. However, the health of the market is not absolute, and there is still room for ambiguity between the two. The land market itself is non-equilibrium in nature, and its development is a dynamic process and is constantly improved with time. Overheating or overcooling of the land market economy does not mean an unhealthy market or even a crisis. Therefore, it is difficult for the evaluation standard of the healthy operation of the land market to be universally applicable. The current residential land market in China can be divided into primary and secondary markets according to the different transaction subjects and levels of flow, where the primary market refers to the market in which the right to use state-owned construction land is ceded to residential land users for a certain period of time [36]. Due to the limitation of data, the residential land market mentioned in this study refers to the primary land market only. In order to objectively evaluate the operation status of the market, this study evaluates the "hot and cold" residential land market conditions, establishes an evaluation system for the operation status of the residential land market, defines the criteria for measuring the "hot and cold" status of the market, evaluates each dimension of the residential land price market in different periods The market operation status is divided into five states: "overheated, slightly hot, healthy, slightly cold, and overcooled" to reveal the operation status of the residential land market in the Beijing-Tianjin-Hebei region and its differentiation characteristics. The results of this study can provide a theoretical basis for land market regulation and the coordinated development of Beijing-Tianjin-Hebei region.

## Data and methods

### Study area

The Beijing-Tianjin-Hebei region is the economic center of the north, including two municipalities directly under the central government of Beijing and Tianjin, as well as 11 prefecture-level cities in Hebei Province, including Shijiazhuang, Tangshan, Baoding, Langfang, Qinhuangdao, Zhangjiakou, Chengde, Cangzhou, Hengshui, Xingtai and Handan, with 174 county-level administrative regions under its jurisdiction. The geographical area is 218,000 square kilometers, accounting for 2.3% of the total area of China. As the political, economic and cultural center of the country, Beijing has significant advantages in policy orientation, resource allocation and industrial structure, and its economic level and competitiveness are much higher than those of other regions. Tianjin, as an important commercial port in northern China, is the core area of international shipping in northern China and is strongly

supported by national policies. Compared with Beijing and Tianjin, Hebei Province has a larger gap in economic level and does not receive as much support in terms of policies as the former two, resulting in unbalanced development in the Beijing-Tianjin-Hebei region. Due to the different geographical location, urban planning, social culture and economic development of different cities, the rhythm of land supply and demand, the structure of land supply, the efficiency of land supply and the degree of land marketization will also be significantly different. As a "barometer" that can reflect the operation of the land market, the spatial and temporal variation of land price is a direct manifestation of the difference in land market conditions.

## Data

In order to precisely analyze the spatial and temporal characteristics of the operation of the residential land market in Beijing, Tianjin and Hebei, this study is based on the administrative division map of the Beijing-Tianjin-Hebei region in 2020, and 174 county-level administrative division units are divided as the research objects. The residential land market data in the study includes land market transaction data and socio-economic data for each county in Beijing, Tianjin and Hebei. Among them, the land market transaction data include land supply plan, total construction land supply, residential land supply, stock residential land use, residential land supply by "auction", supply of land for guaranteed housing, and residential land sale revenue, etc., through the China Land Market Network (https://www.landchina.com/), websites of natural resources bureaus in each city, and field research. Socio-economic data mainly include GDP, disposable income of urban residents, general fiscal budget revenue, year-end urban resident population, price index and other indicators, and data of each indicator are obtained from the 2011–2020 China Urban Statistical Yearbook, Beijing Statistical Yearbook, Tianjin Statistical Yearbook and Hebei Economic Yearbook.

## Methods

This study evaluates the market from the perspective of "hot and cold" and finally classifies the market operation into five states: "overheated, slightly hot, healthy, slightly cold, overcooled", among which, the healthy state only represents a relatively balanced or stable state of land market development, while overheated or overcooled does not It only reflects the size of the market, the speed of development, and the high and low land prices. This study evaluates the state of market operation in four dimensions: the degree of coordination between land price and national economy, the degree of structural equilibrium, the degree of balance between supply and demand, and the degree of market activity.

**Market state index evaluation.** *(1) Indicator system establishment.* The selection of evaluation indexes revolves around the basic attributes of land market such as scale, structure, supply and demand, and price, and fully considers the coordination relationship between market operation and socio-economic development. Through the preliminary selection analysis, this study determines 12 evaluation indicators, which are divided into 4 dimensions such as the coordination degree of residential land price and national economy, structural balance degree, supply and demand balance degree, and market activity degree through cluster analysis to establish the evaluation indicator system. The evaluation index system is shown in Table 1.

*(2) Determination of index weights.* The entropy weighting method is used to determine the weights of residential land market condition evaluation indicators, and the calculation formula is as follows.

**Table 1. Evaluation index system of primary market operation status of residential land.**

| Dimension | Indicator layer | Calculation formula and interpretation | Description of indicators | Weight |
|---|---|---|---|---|
| Coordination degree of land price and national economy $D_1$ | Land-GDP elasticity coefficient ($X_1$) | $$X_{(1,2,3)} = \frac{\sqrt[j-i]{P_j/P_i} - 1}{\sqrt[j-i]{G_j/G_i} - 1}$$ $X_{(1,2,3)}$ represents the land price-GDP elasticity coefficient (or land price-UPDI elasticity coefficient, land price-RPI elasticity coefficient), $P$ denotes the average residential land price level in the county (city, district), $G$ denotes the county (city, district) GDP (or UPDI, LPRPI), and $i$ and $j$ denote the year ($j > i$). | Characterize the coordination of land price and macroeconomic development | 0.1094 |
| | Land price-UPDI elasticity coefficient ($X_2$) | | Characterizing the coordination between land price and resident income | 0.0710 |
| | Land price-RPI elasticity coefficient ($X_3$) | | Characterize the coordination between land price and retail commodity prices | 0.0906 |
| Market structure equilibrium degree $D_2$ | Proportion of residential land supply ($X_4$) | $X_4 = R_{i,j} / T_{i,j}$ <br> $X_4$ represents the proportion of residential land supply, $R_{i,j}$ represents the residential land supply area of the county (city, district) $I$ in year $J$, and $T_{i,j}$ represents the total construction land supply area of the county (city, district) $I$ in year $J$. | Reflect the proportion structure of residential land in the market | 0.1259 |
| | Degree of marketization of residential land supply ($X_5$) | $X_5 = r_{i,j} / R_{i,j}$ <br> $X_5$ represents the degree of marketization of residential land supply, $r_{i,j}$ represents the area of residential land sold by county (city, district) $I$ through bidding, auction, and listing in year $J$, and $r_{i,j}$ represents the total area of residential land supply by county (city, district) $I$ in year $J$. | Reflect the structure of land supply mode in the market | 0.0853 |
| | Stock land use rate ($X_6$) | $X_6 = C_{i,j} / R_{i,j}$ <br> $X_6$ represents the utilization rate of stock land, $C_{i,j}$ represents the stock land area of residential land supplied by county (city, district) $I$ in year $J$, and $R_{i,j}$ represents the total residential land supply area of the county (city, district) $I$ in year $J$. | Reflect the proportion structure of stock land in the market | 0.0893 |
| Market supply and demand balance $D_3$ | Supply rate of residential land ($X_7$) | $X_7 = R_{i,j} / J_{i,j}$ <br> $X_7$ represents the supply rate of residential land, $J_{i,j}$ represents the planned area of residential land supply of county (city, district) $I$ in year $J$, and $R_{i,j}$ represents the total area of residential land supply of county (city, district) $I$ in year $J$. | Measuring the balance between residential land supply and planned demand | 0.0800 |
| | Per capita residential land supply area ($X_8$) | $X_8 = R_{i,j} / P_{i,j}$ <br> $X_8$ represents the per capita residential land supply area, $P_{i,j}$ represents the urban permanent population size of the county (city, district) $I$ in year $J$, and $R_{i,j}$ represents the total residential land supply area of the county (city, district) $I$ in year $J$. | Measuring the balance between residential land supply and population demand | 0.0965 |
| | Proportion of land for social indemnificatory housing ($X_9$) | $X_9 = B_{i,j} / R_{i,j}$ <br> $X_9$ represents the proportion of land for social indemnificatory housing, $B_{i,j}$ represents the supply area of land for social indemnificatory housing in the county (city, district) $I$ in year $J$, and $R_{i,j}$ represents the total supply area of residential land in county (city, district) $I$ in year $J$. | Measuring the supply and demand of land for social indemnificatory housing | 0.0665 |
| Market activity $D_4$ | Financial contribution of residential land transfer ($X_{10}$) | $X_{10} = N_{i,j} / (N_{i,j} + M_{i,j})$ <br> $X_{10}$ represents the financial contribution of residential land transfer, $N_{i,j}$ represents the residential land transfer income of county (city, district) $I$ in year $J$, and $M_{i,j}$ represents the local general public budget income of county (city, district) $I$ in year $J$. | Reflect the degree of dependence on land finance and market heat | 0.0504 |
| | Number of residential land transactions ($X_{11}$) | —— | Reflect the number of residential land transactions | 0.0548 |
| | Residential land transaction area ($X_{12}$) | —— | Reflect the transaction area of residential land | 0.0803 |

① Construct a decision matrix of $n$ evaluation indexes of the $m$ evaluation unit.

$$A = \left(X_{ij}\right)_{m \times n} (i = 1, 2, \ldots, m; j = l, 2, \ldots, n) \tag{1}$$

② The dimensionless processing of the indicator values yields the normalized judgment matrix.

$$R = \left( r_{ij} \right)_{m \times n} \tag{2}$$

The "minimum-maximum normalization" method is used to linearly transform the original index data. The corresponding normalization equations for positive and negative indicators are:

$$c_{ij} = \frac{r_{ij} - r_i^{min}}{r_i^{max} - r_i^{min}}, c_{ij} = \frac{r_i^{max} - r_{ij}}{r_i^{max} - r_i^{min}}$$
$$(i = 1, 2, \ldots, m; j = l, 2, \ldots, n) \tag{3}$$

Where $c_{ij}$ is the standardized value of the $j$th evaluation index of evaluation unit $i$.

③ The weight of each index value is calculated to further determine the information entropy of each index.

$$p_{ij} = c_{ij} / \sum_{i=1}^{m} c_{ij} \quad (i = 1, 2, \ldots, m; j = 1, 2, \ldots, n) \tag{4}$$

$$e_j = -k \sum_{i=1}^{m} p_{ij} \ln\left( p_{ij} \right), k = 1/ln(n) \tag{5}$$

Where, $P_{ij}$ is the proportion of the $i$th evaluation object of the $j$th index to the index, and $ej$ is the information entropy of the $j$th index.

④ Calculate the information utility value of the jth index, and further calculate the weight of each index.

$$d_j = 1 - e_j \tag{6}$$

$$F_i = \sum_{j=1}^{n} D_{ij} \times W_{ij} \tag{7}$$

Where, $d_j$ is the information utility value of the *jth* index, and $w_j$ is the weight value of the *jth* index.

The evaluation index system of residential land market status is as follows:

*(3) Evaluation method.* The "minimum-maximum standardization" method was used to standardize the original index data, and the comprehensive index method was used to calculate the indices of each dimension and the total market state score of each evaluation unit.

$$D_{ij} = \sum_{j=1}^{m} c_{ij} \times w_{ij} \tag{8}$$

$$F_i = \sum_{j=1}^{n} D_{ij} \times W_{ij} \tag{9}$$

where $D_{ij}$ is the $j$th evaluation dimension index of evaluation unit $i$, $c_{ij}$ is the standardized value of the $j$th evaluation index of evaluation unit $i$, $w_{ij}$ is the weight of the $j$th evaluation

index of evaluation unit *i*, $F_{ij}$ is the total market state score of evaluation unit *i*, and $W_{ij}$ is the weight of the *j*th evaluation dimension of evaluation unit *i*.

**Definition of "hot and cold" status criteria.** The evaluation indexes of this study are all positive indicators, and the larger the indexes of each dimension and the comprehensive index of market status represent the higher the hotness of the residential land market development status. To evaluate the "hot and cold" of each dimension and the overall operation of the market, statistical analysis of each dimension index and the comprehensive index of the market state is required, and the five states of " overheated, slightly hot, healthy, slightly cold, overcooled" are formulated through the K-S test and the 3σ method in statistics. Definition criteria.

*(1) K-S test analysis*. The premise of the 3σ interval division method is that the evaluation data have a normal distribution pattern. We first normalize each dimensional index and market state comprehensive score by using LOG transformation method to make them obey normal distribution, and then test the transformed data by K-S test method. The K-S statistics of the sample data are calculated by SPSS software, and the corresponding companion probability values are given according to the K-S distribution table. If the companion probability is less than or equal to the significance level a (0.05), the null hypothesis H0 should be rejected, and the sample data are considered to be significantly different from the normal distribution; if the companion probability is greater than the significance level, the null hypothesis H0 cannot be rejected and the overall sample data is considered to have normally distributed characteristics.

*(2) 3σ method principle*. By normalization, each evaluation index X obeys the normal distribution: X ~ N (μ, σ2), then the span of the health status interval of each index is xi. According to the principle of normal distribution, the closer the data are to the mean, the higher the probability. The probability that the values are distributed in (μ-σ, μ+σ) is 0.6827, the probability that the values are distributed in (μ-2σ, μ+2σ) is 0.9545, and the probability that the values are distributed in (μ-3σ, μ+3σ) is 0.9973. Therefore, for the indicators that meet such characteristics, we can reflect whether the indicator data are normal or not based on the standard deviation multiple of deviation from the mean, and take the (μ-σ, μ+σ) interval as the reference range of normal distribution of indicators, and select the principle of interval division based on the number of evaluation units between the normal distribution range.

If the difference between the values of indicators of evaluation units is small, most of them are located in the reference range of normal distribution, that is, the number of indicators distributed in the (μ-σ, μ+σ) interval is ≥2/3. At this time, the interval of 2/3 number of indicators is taken as the interval of health status. The mathematical expression is: P{X-μ<xi} = 2/3, that is, assuming that the indicator data set obeys the standard normal-terrestrial distribution N (0, 1), the probability of indicator X falling into the healthy interval span (-xi, xi) is 2/3, then P (-xi < X< xi) = F(-xi)—F(xi) = 2 × F(xi) − 1 = 2/3, which gives F(xi) = 0.8333.

From the normal distribution table, xi = 0.97σ, then the operating state interval of the indicator X is: (μ-0.97σ, μ+0.97σ). The corresponding overcooled, slightly cold, slightly hot, and overheated intervals are calculated as (∞, μ-1.94σ], (μ-1.94σ, μ-0.97σ], [μ+0.97σ, μ+1.94σ), and [μ+1.94σ, +∞), respectively.

## Results

### Results of residential land market index measurement by city

The comprehensive index method was used to calculate the residential land market dimension index and the comprehensive market state index for each county. Taking the cities as units, the average value of the residential land market status index of all of the counties within the jurisdiction of each city was calculated as the city's residential land market status index.

**Residential land market dimension index by City.**   The land price and national economic coordination index (D1) is shown in Fig 1. During 2011–2015 and 2016–2020, the land price and national economic coordination indexes of each city ranged from 0.028 to 0.058 and from 0.027 to 0.118, respectively. Beijing and Zhangjiakou were always at a higher level, with D1 values greater than 0.05 during both time periods. Qinhuangdao, Xingtai, and Chengde were at a lower level, with D1 values less than 0.04 during both time periods. This indicates that the residential land price levels in Beijing and Zhangjiakou were significantly high relative to the economic development indicators such as the GDP, urban residents' disposable income level (UPDI), and land price-retail price index (LPRPI), while the residential land price levels in Qinhuangdao, Xingtai, and Chengde were relatively low. The D1 values in Tianjin, Langfang, Cangzhou, and Baoding increased significantly from 2016 to 2020, indicating that the residential land price levels in these cities increased at a significantly faster rate relative to the national economic growth rate.

The market structural equilibrium index (D2) is shown in Fig 1. The D2 values ranged from 0.106 to 0.166 and from 0.108 to 0.175 in each city during 2011–2015 and 2016–2020, respectively. Langfang had the highest level, with D2 values greater than 0.16 during both time periods. Tianjin had the lowest level, with D2 values less than 0.11 during both time periods. During 2016–2020, although the indexes of this dimension increased in Langfang and Zhangjiakou, these indexes were relatively similar in most of the cities and remained basically stable over time. This indicates that most of the cities had similar characteristics in terms of the proportion of the residential land supply, the degree of marketization, and the usage structure, and the changes were all relatively stable.

The market supply and demand balance index (D3) is shown in Fig 1. The D3 values in each city ranged from 0.027 to 0.067 and from 0.028 to 0.060 during 2011–2015 and 2016–

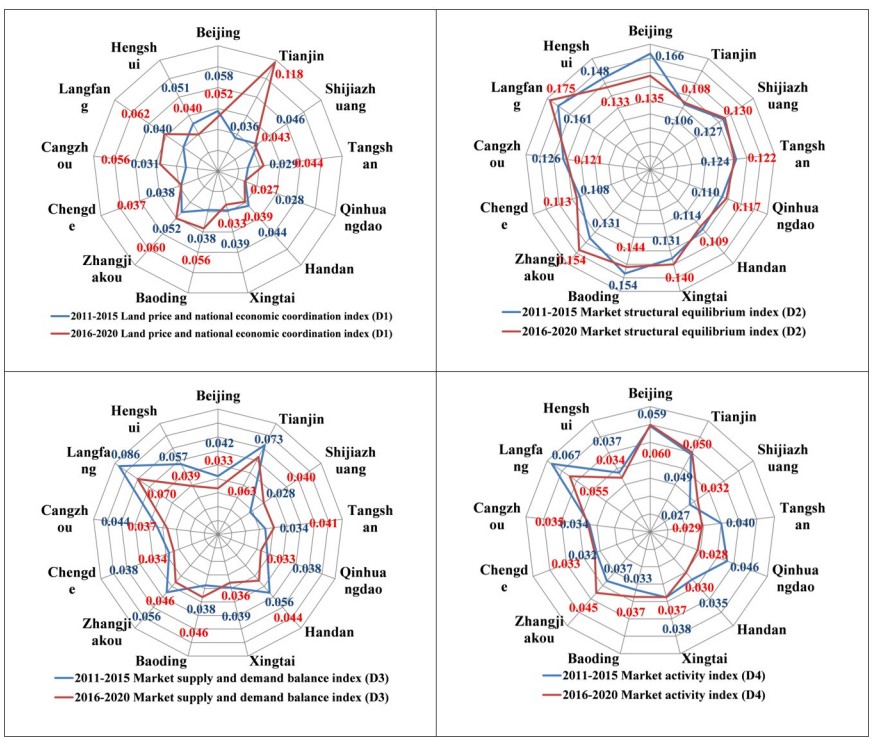

**Fig 1. Spider web diagram of the residential land market dimension indexes of city.**

2020, respectively. Tianjin and Langfang were always at a high level, with D3 values greater than 0.06 during both time periods. Qinhuangdao, Xingtai, and Chengde were always at a low level, with D3 values less than 0.04 during both time periods. During 2016–2020, except for Shijiazhuang, Tangshan, and Baoding, the index of this dimension decreased significantly in all the other cities. This indicates that there were obvious differences among the cities in terms of the land supply rate of the residential land, the supply area of the residential land per capita, and the proportion of land for guaranteed housing; in most of the cities, these indexes decreased with time.

The market activity index (D4) is shown in Fig 1. The D4 values in each city ranged from 0.027 to 0.067 and from 0.028 to 0.060 during 2011–2015 and 2016–2020, respectively. Beijing and Langfang were always at a high level, with D4 values greater than 0.055 during both time periods, indicating that these two cities were relatively high in terms of the number of residential land transactions and the contribution of the fiscal revenue. Shijiazhuang, Xingtai, Chengde, and Cangzhou were always at a lower level, with D4 values less than 0.04 during both time periods. The gap between the cities in terms of this index narrowed overall after 2016, with this index increasing in Zhangjiakou, Baoding, and Shijiazhuang and significantly decreasing in Langfang, Tangshan, Qinhuangdao, and Handan.

The above results show that Beijing's index levels were generally high in all dimensions, except for the market supply and demand balance index, but the land price and national economy coordination index and the market supply and demand balance index exhibited decreasing trends. In Tianjin, except for the market structural equilibrium index, the indexes of the other dimensions were also at a high level, and the index of the coordination between the land price and the national economy increased significantly. In Hebei, most of the dimensional indexes were generally highest in Langfang, followed by Zhangjiakou, Baoding, Cangzhou, and other cities near Beijing and Tianjin; however, the dimensional indexes were at a low level and exhibited decreasing trends in Qinhuangdao, Chengde, Xingtai, Handan, and other marginal cities. It can be seen that the operation status of each dimension of the urban residential land market was related to the location conditions, and the higher socio-economic development level and the strong demand for land resource elements in Beijing and Tianjin played a radiating and driving role for the neighboring cities; however, the connection between these two cities and the cities on the edge of the region was relatively weak, and the spatial agglomeration effect of the development of the residential land market in the entire region was greater than the diffusion effect.

**Composite index of residential land market by city.** As is shown in Fig 2, the comprehensive index of the residential land market status of each city ranged from 0.2156 to 0.3544 and from 0.2057 to 0.3624 during 2011–2015 and 2016–2020, respectively. Langfang had the highest market state composite index, with values greater than 0.35 during both time periods. Tianjin, Beijing, Zhangjiakou, and Baoding were at a higher level, with composite index values greater than 0.26 during both time periods. Hengshui, Cangzhou, Xingtai, Handan, Qinhuangdao, Tangshan, and Shijiazhuang had moderate composite index levels, with composite index values of 0.22–0.25 during both time periods. Chengde had the lowest composite index, with composite index values less than 0.22 during both time periods. The differences between the comprehensive index levels of most of the cities were not large, and the changes in the index before and after 2016 were basically stable, indicating that the residential land market in most of the cities was running in a relatively similar state, and the overall residential land market in the entire region was running in a relatively stable state. By comparing the two time periods, it was found that from 2016 to 2020, the market state indexes decreased significantly in Beijing and Hengshui and increased significantly in Tianjin, Zhangjiakou, Baoding, Cangzhou, Shijiazhuang, and other cities adjacent to Beijing. This indicates that the development of the

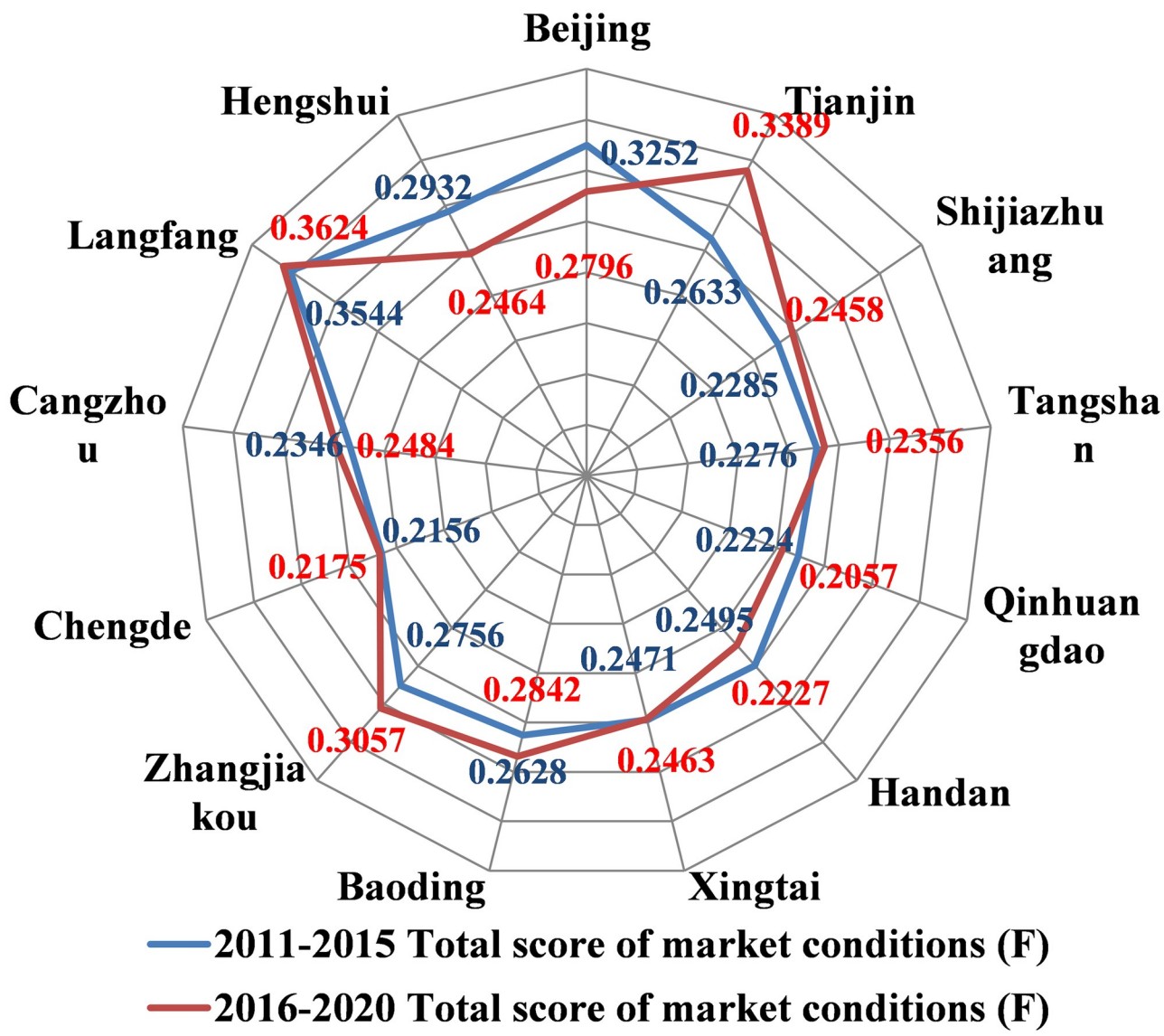

**Fig 2. Spider web diagram of the residential land market status (F) composite index of city.**

residential land market in Beijing slowed down, but the development of the residential land market in its neighboring cities increased. In addition, the residential land market in the Beijing-Tianjin ring area exhibited a synergistic development trend.

### The pattern of "hot and cold" residential land market in the county scale

**Results of the normal distributivity test of the evaluation data.** Sample data with normal distribution characteristics is a prerequisite for applying the 3σ rule analysis. In this study, based on the index evaluation data of 174 counties in Beijing, Tianjin and Hebei, the market dimension index and market state composite index of the two time periods are LOG transformed to make them conform to the law of normal distribution, and verified by the one-sample K-S test method. The results are shown in Table 2.

**Table 2. Table of K-S test results for each evaluation data.**

| Evaluation Data | 2011–2015 | | | 2016–2020 | | |
|---|---|---|---|---|---|---|
| | Conversion Methods | Kolmogorov-Smirnov Z | Asymp. Sig. (2-tailed) | Conversion Methods | Kolmogorov -Smirnov Z | Asymp. Sig. (2-tailed) |
| D1 | LOG | .914 | .374 | LOG | .522 | .948 |
| D2 | LOG | .806 | .535 | LOG | .753 | .621 |
| D3 | LOG | .574 | .897 | LOG | .876 | .426 |
| D4 | SQRT | .834 | .490 | SQRT | .635 | .815 |
| F | LOG | .678 | .747 | LOG | .734 | .654 |

As can be seen from Table 2, Asymp. Sig.(2-tailed), i.e., the probability of companionship, is greater than the general significance level (0.05), and the original hypothesis of H0 is accepted, indicating that each dimensional index and market state composite index have normal distribution characteristics, and the 3σ method can be used to classify the "hot and cold" state intervals.

**Market "hot and cold" state interval classification results.** In order to make the operation of the residential land market in the two time periods from 2011 to 2015 and 2016 to 2020 comparable, this study adopts a unified market state interval division standard. The analysis of the data in Table 2 shows that the evaluation data from 2011 to 2015 have a strong overall normal distribution pattern and the 3σ method works better, so the market state interval division is conducted only for the evaluation data from 2011 to 2015, and it is used as the unified market state interval standard. SPSS software was used to statistically analyze the indices of each dimension and the comprehensive index of market operation status from 2011 to 2015 to grasp the overall distribution of each data set, and the interval division method was selected based on the number of evaluation units located in the normal distribution range (μ-σ, μ+σ). The statistical analysis is presented in Table 3.

As can be seen from Table 3, the sample units in the study area are 174 counties, and 2/3 or more of the sample numbers of each evaluation data are located in the normal distribution range (μ-σ, μ+σ), which indicates that the overall operating condition of the primary market of residential land in the whole region is good and in line with the principle of 3σ method. Therefore, 2/3 of the evaluation sample size is taken as the healthy condition interval, and accordingly the overcooled, slightly cold, healthy, slightly hot, and overheated intervals are calculated as (∞, μ-1.94σ], (μ-1.94σ, μ-0.97σ], (μ-0.97σ, μ+0.97σ), [μ+0.97σ, μ+1.94σ), [μ+1.94σ, +∞), respectively. The statistical values of each index were compared with the five running state intervals in turn, and the converted evaluation data were reduced to the original values of the

**Table 3. Statistical analysis results of each evaluation index from 2011 to 2015.**

| Statistical quantities | D1 | D2 | D3 | D4 | F |
|---|---|---|---|---|---|
| Average value | 0.0414 | 0.1326 | 0.0466 | 0.0389 | 0.2594 |
| Number of samples | 174 | 174 | 174 | 174 | 174 |
| Standard deviation | 0.0217 | 0.0320 | 0.0282 | 0.0219 | 0.0628 |
| Minimal value | 0.0039 | 0.0585 | 0.0077 | 0.0000 | 0.1462 |
| Maximum value | 0.1772 | 0.2143 | 0.1411 | 0.1215 | 0.4797 |
| full distance | 0.1734 | 0.1557 | 0.1334 | 0.1215 | 0.3335 |
| μ-σ | 0.0197 | 0.1005 | 0.0184 | 0.0171 | 0.1966 |
| μ+σ | 0.0630 | 0.1646 | 0.0748 | 0.0608 | 0.3222 |
| Number of normal samples | 129 | 122 | 119 | 124 | 119 |

**Table 4. Each dimension and the overall market operation state "hot and cold" interval table.**

| Indicators | Overcooled | Slightly cold | Health | Slightly hot | Overheated |
|---|---|---|---|---|---|
| D1 | (-∞, 0.0153) | [0.0153, 0.0203) | [0.0203, 0.0592) | [0.0592, 0.1028) | [0.1028, ∞) |
| D2 | (-∞, 0.0787) | [0.0787, 0.1009) | [0.1009, 0.1650) | [0.1650, 0.2007) | [0.2007, ∞) |
| D3 | (-∞, 0.0107) | [0.0107, 0.0202) | [0.0202, 0.0712) | [0.0712, 0.1037) | [0.1037, ∞) |
| D4 | (-∞, 0.0110) | [0.0110, 0.0203) | [0.0203, 0.0601) | [0.0601, 0.0881) | [0.0881, ∞) |
| F | (-∞, 0.1635) | [0.1635, 0.2005) | [0.2005, 0.3168) | [0.3168, 0.4059) | [0.4059, ∞) |

indexes and the corresponding belonging intervals were found out, and finally a table of the running state intervals of each evaluation data was formed, which is shown in Table 4.

**Residential land market "hot and cold" status pattern.** According to the measured dimensional indices (D1~D4), the dimensional indices of residential land market in each town were divided into five states: overcooled, slightly cold, healthy, slightly hot, overheated, and were displayed in graphs with the help of ArcGIS 10.2 software, and the results are shown in Table 5.

*(1) Status of each dimension of the residential land market.* The land price and the coordination degree of the national economy (D1) are under cooling, slightly cold, healthy, slightly hot and overheated. During 2011~2015, the proportion of the county seat is 2.87%, 10.34%, 72.99%, 10.92% and 2.87% respectively. The proportion of the county seat that is slightly cold (overcooled, slightly cold) and slightly hot (slightly hot, overheated) is relatively balanced, and most of the county seats are in a healthy state. The proportions of county towns in healthy states from 2016 to 2020 were 0.57%, 5.17%, 69.54%, 19.54%, and 5.17%, with the healthy state towns decreasing but still accounting the majority of the total towns. The spatial pattern is shown in Fig 3. From 2011 to 2015, the cold and hot state counties were distributed in each city and exhibited local clustering characteristics. After 2016, the number of hot counties increased significantly, with 10.92% more than before 2015. They were mainly clustered in Tianjin, Zhangjiakou, Baoding, Cangzhou, and other urban jurisdictions. The residential land price level in some of the counties accelerated significantly relative to the national economic rate of increase. This was most obvious in the counties around Beijing and Tianjin.

The proportion of county towns in the five states of market structure equilibrium (D2) from 2011 to 2015 is 2.87%, 11.49%, 69.54%, 13.79% and 2.30% respectively. From 2016 to 2020, the proportion was 3.45%, 11.49%, 70.11%, 12.64% and 2.30% respectively. The healthy counties accounted for the majority during both time periods, and the numbers of counties in the other states remained relatively stable. The pattern is shown in Fig 4. From 2011 to 2015, the hot state counties were mainly concentrated in the Beijing and Baoding jurisdictions. From 2016 to 2020, the hotness of the counties in the Beijing and Baoding jurisdictions decreased significantly and most of the counties changed to healthy state, while the hotness of

**Table 5. Structure of the number of towns for each dimensional index state.**

| Market Status | Percentage of counties in 2011–2015 | | | | Percentage of County 2016–2020 | | | |
|---|---|---|---|---|---|---|---|---|
| | D1 | D2 | D3 | D4 | D1 | D2 | D3 | D4 |
| Overcooled | 2.87% | 2.87% | 2.87% | 3.45% | 0.57% | 3.45% | 2.30% | 4.02% |
| Slightly cold | 10.34% | 11.49% | 9.20% | 10.92% | 5.17% | 11.49% | 13.22% | 14.94% |
| Health | 72.99% | 69.54% | 68.97% | 69.54% | 69.54% | 70.11% | 69.54% | 66.67% |
| Slightly hot | 10.92% | 13.79% | 13.79% | 10.34% | 19.54% | 12.64% | 13.22% | 9.77% |
| Overheated | 2.87% | 2.30% | 5.17% | 5.75% | 5.17% | 2.30% | 1.72% | 4.60% |

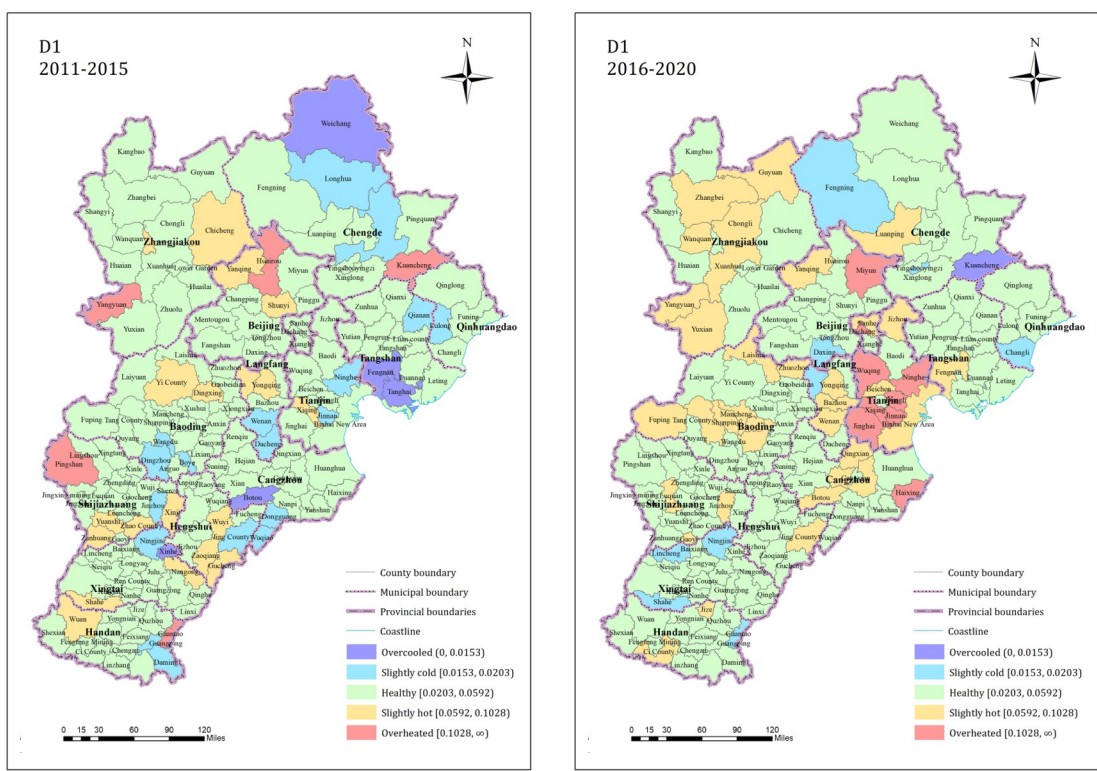

**Fig 3. D1 dimension residential land market index state spatial pattern.**

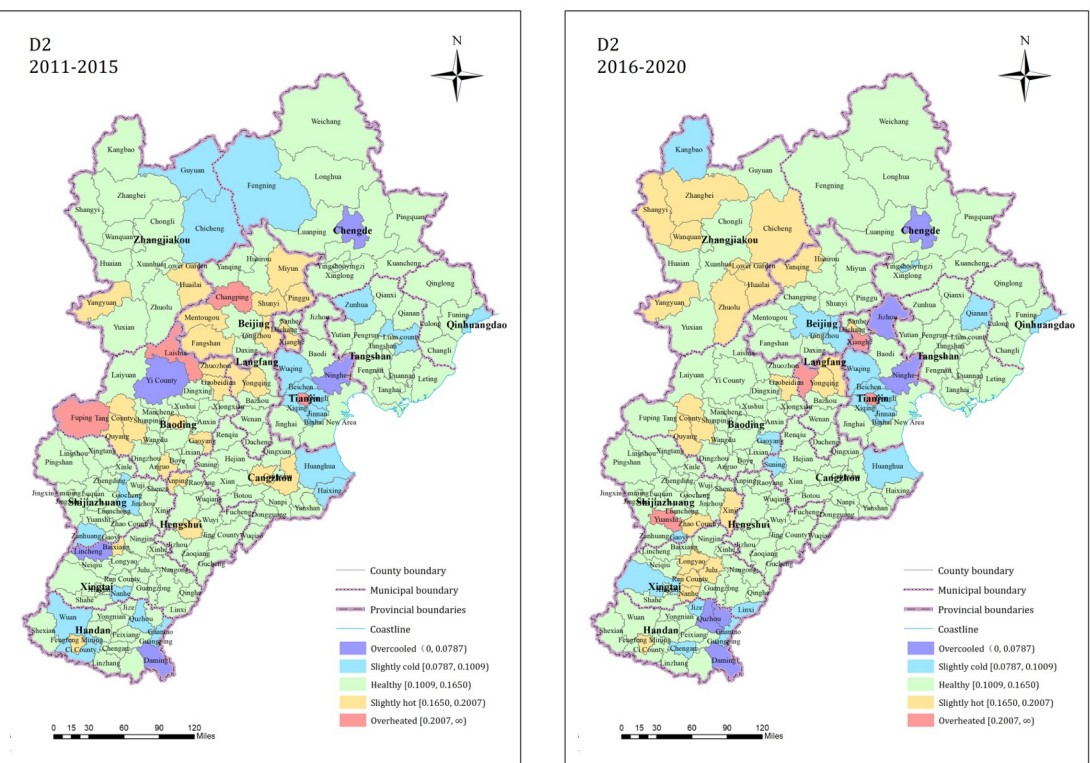

**Fig 4. D2 dimension residential land market index state spatial pattern.**

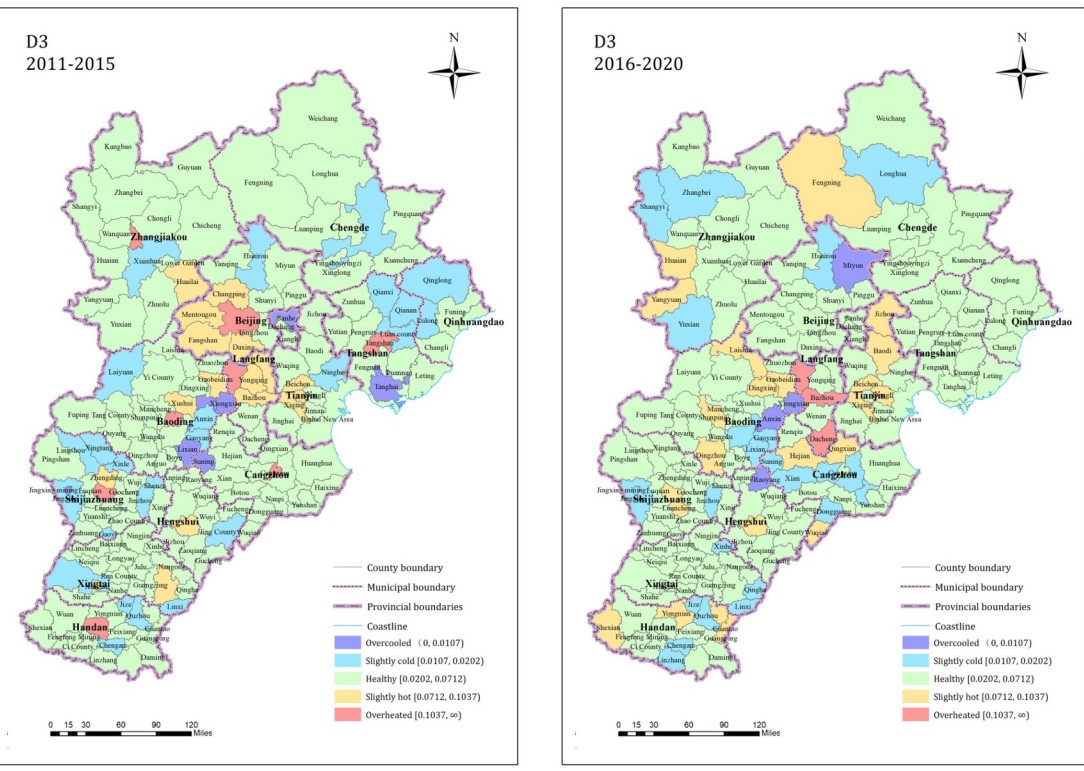

**Fig 5. D3 dimension residential land market index state spatial pattern.**

the counties in the Zhangjiakou City jurisdiction generally increases. During the two time periods, the hot and cold state counties always exhibited local small-scale clustering.

The proportions of counties with the market supply and demand balance (D3) in the overcooled, slightly cold, healthy, slightly hot, and overheated states during 2011–2015 were 2.87%, 9.20%, 68.97%, 13.79%, and 5.17%, respectively, and most counties were in a healthy state. During 2016–2020, the proportion of counties in these states were 2.30%, 13.22%, 69.54%, 13.22%, and 1.72%; the number of healthy and cold state counties increased, but the overall state structure was stable. The pattern is shown in Fig 5. The hot and cold state counties exhibited local clustering characteristics. The number of counties with healthy market supply-demand balance in the Beijing and Tianjin jurisdictions increased, and the increase in the counties with a cold status was mainly concentrated in Chengde, Zhangjiakou, Baoding, and Cangzhou, and other cities around the Beijing and Tianjin jurisdictions.

The proportions of counties with market activity (D4) index in the overcooled, slightly cold, healthy, slightly hot, and overheated states during 2011 to 2015 were 3.45%, 10.92%, 69.54%, 10.34%, and 5.75%, respectively; the proportions from 2016 to 2020 were 4.02%, 14.94%, 66.67%, 9.77%, and 4.60%, respectively. During both time periods, most of the counties were in a healthy state. The proportion of counties in cooler states increased after 2016, and the proportions of counties in healthy and hotter states decreased slightly. This pattern is shown in Fig 6. The central counties of each city jurisdiction were significantly hotter than the other counties. Beijing, the Langfang jurisdictional counties, and the surrounding counties were always more obviously partially in a hot state and exhibited obvious clustering characteristics. This indicates that the market activity in the counties with good location conditions and

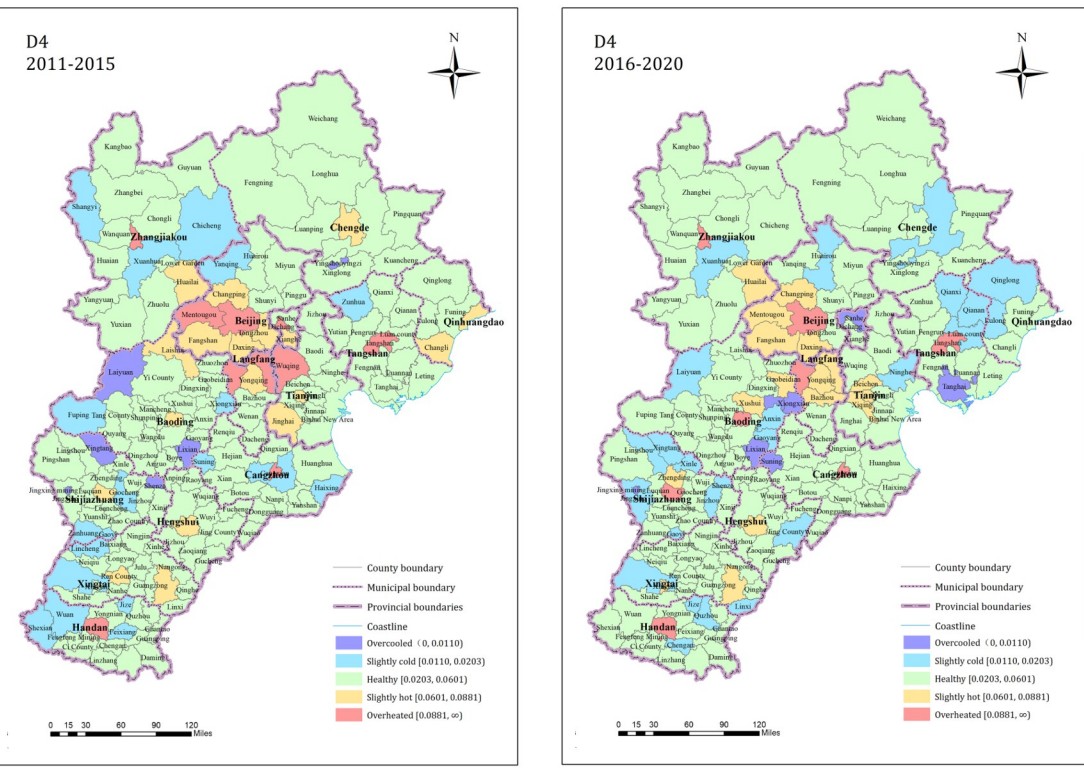

**Fig 6. D4 dimension residential land market index state spatial pattern.**

a high economic level was generally high. The increase in the cold state counties mainly occurred in the southern and northeastern regions of Hebei and the edge of the county, exhibiting a small range of clustering.

*(2) Overall operation of the residential land market.* The comprehensive index of residential land market condition (F) was calculated by the index of each dimension and its weight. According to the evaluation data state interval table, the overall operation condition of the regional residential land market is divided into five states: overcooled, slightly cold, healthy, slightly hot and overheated, and is displayed as a map with the help of ArcGIS 10.2 software, and the results are as follows.

As is shown in Table 6, the proportions of the counties in the overcooled, slightly cooled, healthy, slightly hot, and overheated states in terms of the residential land market in the Beijing-Tianjin-Hebei region from 2011 to 2015 were 1.72%, 13.79%, 67.24%, 13.79%, and 3.45%, respectively. The proportions of the counties in these states from 2016 to 2020 were 2.87%,

**Table 6. Residential land market overall operation status structure table.**

| Market Status | Percentage of towns in 2011–2015 | Percentage of towns in 2016–2020 |
| --- | --- | --- |
| Overcooled | 1.72% | 2.87% |
| Slightly cold | 13.79% | 12.64% |
| Health | 67.24% | 64.94% |
| Slightly hot | 13.79% | 17.24% |
| Overheated | 3.45% | 2.30% |

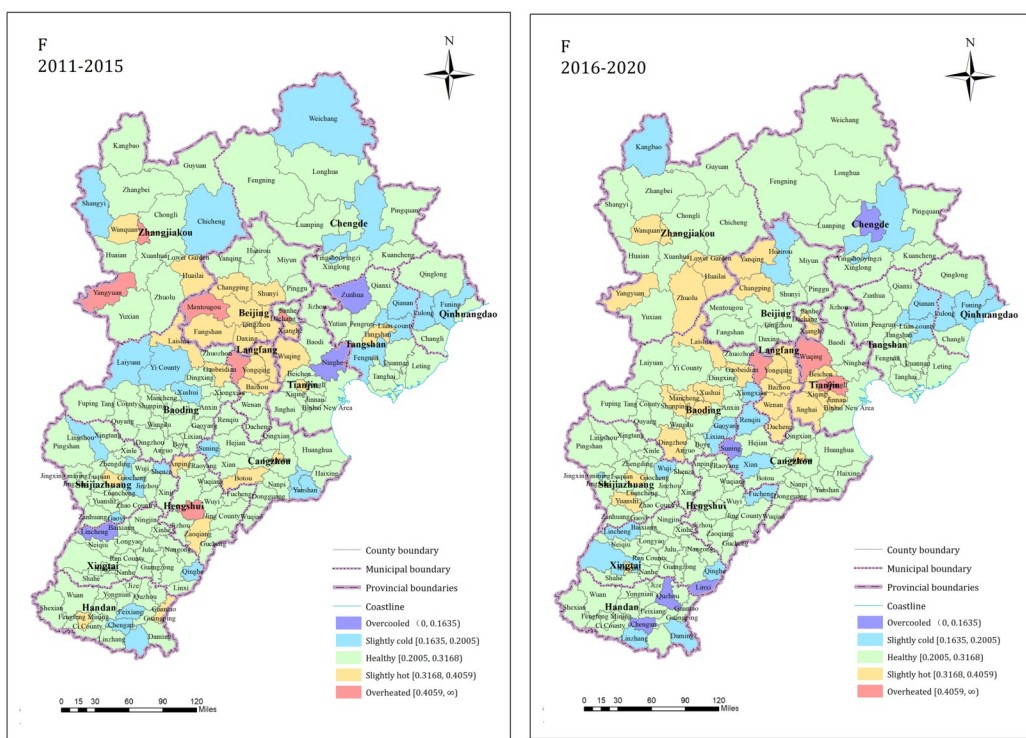

**Fig 7. Overall operating state pattern of the residential land market.**

12.64%, 64.94%, 17.24%, and 2.30%. The healthy state counties accounted for the majority during both time periods, and the numbers of towns in other states remained relatively stable. The proportion of counties in a healthy and cold state decreased slightly, and the proportion of counties in a hot state increased slightly in the entire region from 2016 to 2020. The spatial pattern is shown in Fig 7. During 2011–2015, the number of hot-leaning counties in the Beijing and Langfang jurisdictions was high and exhibited significant clustering. During 2016–2020, the hotness of the counties in the Beijing jurisdictions decreased and gradually changed to a healthy status; however, the market hotness in the surrounding areas increased significantly. In particular, the hotness of the counties near Beijing such as Langfang, Tianjin, Zhangjiakou, and Baoding increased significantly. The cooler counties in the region always exhibited a more fragmented distribution but exhibited local small-scale clustering. With the rapid development of urbanization, the overall running speed of the residential land market in the entire region accelerated. This was the most obvious in the areas around Beijing. The development speed of the residential land market in the Beijing's jurisdictions decreased significantly, with most of the jurisdictions returning to a healthy state. The number of towns in Tianjin with an overall hot running state in the market increased significantly and the development speed accelerated.

## Discussion

### Comparison of the state of the residential land market in Beijing, Tianjin and Hebei and policy recommendations

Beijing is the political and cultural center of China, and Tianjin is the core city of China's third growth pole—the Bohai Sea Economic Belt. These two cities are characterized by large

population concentration, dense industrial clusters and fast economic development. Among the indicators of the residential land market in these two cities, the area of residential land transactions, land revenue from residential land and the degree of land marketization are significantly higher than those of other cities, and the residential land market has always been more active, with high urban land prices and significant increases. However, compared with the national economic development level, the residential land price level in Beijing and Tianjin is obviously high, which is also accompanied by the "big city disease", which is not conducive to the stable and healthy development of the real estate market in the two cities. Therefore, the three-dimensional indicators of residential land price coordination with national economy, market supply and demand balance and structural balance will become the main factors affecting the healthy development of residential land market in Beijing and Tianjin, and further measures should be taken in terms of stabilizing land price, balancing supply and demand, and optimizing supply structure to ensure the healthy development of residential land market in the two cities.

Among the cities in Hebei Province, Langfang, Zhangjiakou, Baoding, and other cities around Beijing have clear location advantages, and the residential land market in these cities was in the forefront in Hebei Province in terms of each dimensional index and comprehensive index. During 2016–2020, these three cities were influenced by the favorable policies for Beijing-Tianjin-Hebei synergistic development, and the residential land market developed rapidly. All the dimensional and comprehensive status indexes exhibited increasing trends. Langfang, Zhangjiakou, and Baoding have formed new growth poles for the development of the residential land market in Beijing-Tianjin-Hebei, and they can promote the integrated development of the land market in the entire region in the future. However, the southernmost cities (Handan and Xingtai) and northeastern cities (Qinhuangdao and Chengde) in the Beijing-Tianjin-Hebei region have disadvantageous locations, relatively weak traffic accessibility, a limited influenced by the Beijing-Tianjin radiation, relatively low allocation of land market factors, and low residential land market index and comprehensive index values compared to the rest of Hebei Province, which will limit the coordinated development of the Beijing-Tianjin-Hebei land market. Therefore, we should fully implement the city-based policies and maintain the stability of the land market in each city in the region. For cities with obvious location advantages and a hot land market, we should not only guide and supervise the healthy development of the land market in such cities, i.e., diverting the exuberant land demand in the hot cities, but also prevent the risk of the creation of bubbles due to the rapid increase in land prices. For cities with poor location conditions and a cold land market, we should focus on increasing support, steadily promoting infrastructure construction, deepening the implementation of the policy of de-stocking, and promoting healthy operation of the land market.

## County residential land market status pattern characteristics and policy recommendations

The proportion of counties with a healthy status in terms of all four dimensions and the overall market status reached 64% or higher, and the entire regional residential land market was operating well. From 2016 to 2020, the structural balance, supply and demand balance, market activity, and heat index generally decreased, and for the counties in a cold state, these factors increased by 0.58%, 3.45%, and 4.59%. The number of counties with hot land price and national economy coordination increased significantly (by 10.92%), and the number of counties with a hot market status index increased by 2.3%. After 2016, the growth rate of the residential land prices in most of the counties accelerated significantly compared to the growth rate of the national economy (e.g., the GDP, UPDI, and LPRPI), and the real estate prices

became detached from the economic growth and the residents' income level. At present, the government should adhere to the regulation and control policy of stabilizing land prices, stabilizing housing prices, and stabilizing expectations. In addition, new development models should be actively explored, such as implementing rent-to-own situations and promoting the construction of subsidized housing, to support the commercial housing market and to better meet the reasonable housing needs of homebuyers. This would promote a virtuous cycle and healthy development of the residential land market and the real estate market.

In terms of the distribution pattern, both the four dimensions and the overall market operation state exhibited local clustering features in both the hot and cold counties, and most of these parameters were not consistent with the municipal administrative boundaries, such as Beijing, Tianjin, Langfang, and Baoding. This indicates that the residential land market status of the nearby counties was closely related to each other, and the residential land market operation status of the counties was more influenced by the nearby counties than by the radiation from the urban center where they were located. The hot counties were mainly clustered around Beijing and Tianjin, and the clustering phenomenon of the hot counties in the region became increasingly obvious after 2016. To a small extent, the cold counties were mostly clustered in the peripheral areas such as the southern and northeastern parts of the region, and the phenomenon of a cold residential land market in the peripheral areas was still obvious after 2016. In view of the above characteristics, in order to realize the integrated development of the regional residential land market, it is suggested to formulate regulatory measures in two aspects: First, through local linkage and integration, regional linkage development from local to overall should be achieved, spatial planning and industrial planning should be implemented for each city immediately, the planning of spatial resources and other resources should be unified, the market competition rules and control measures should be developed in a unified manner, and an industrial cooperation platform should be constructed to build city-county cities with a reasonable division of labor in the residential land market synergy system and to maximize the land use efficiency. Second, the construction of secondary urban centers and central counties should be supported, Beijing's non-capital core functions should be decentralized, material and financial resources should be concentrated within the urban agglomeration, production factors should be concentrated in the regional central cities and in key counties, the comprehensive strength of the other cities should be improved, especially the peripheral cities, and an urban land market system with a single strong core, several supporting growth poles, and efficient allocation of land factors should be formed.

## Conclusions

In this study, 174 counties in the Beijing-Tianjin-Hebei region were taken as the research objects, land market transaction data from 2011 to 2020 were used, 12 indicators that are highly relevant to the residential land market were selected, the comprehensive index method was used to systematically evaluate the residential land market operation index in four dimensions (i.e., the coordination between land price and the national economy, the balance between supply and demand, structural equilibrium, and market activity), and the residential land market operation index was evaluated using the 3σ rule of the normal distribution. Using the 3σ rule of the normal distribution, the residential land markets in the 174 counties were classified into five operation states, i.e., overcooled, slightly cold, healthy, slightly hot, and overheated, and the distribution pattern of the residential land market states was determined on the city and county scales. The conclusions of this study are as follows.

(1) At the municipal scale: ①Among the urban clusters, Beijing's residential land price coordination with the national economy, market structural equilibrium, market activity index, and market state composite index were all in the middle to upper levels, while the market supply-demand balance index was relatively low and exhibited a decreasing trend. The balance of the market supply and demand may become a constraint affecting the healthy development of the residential land market in Beijing in the future. ② Tianjin's residential land price coordination with the national economy, market supply and demand balance, market activity index, and market state composite index were at a high level, while the market structural equilibrium index was relatively low. In Tianjin, the land price and national economy coordination index increased significantly after 2016. The coordination of the residential land price and the national economy and the market structural equilibrium may become constraining factors affecting the healthy development of the residential land market in Tianjin in the future. ③ Among the cities in Hebei Province, the index of each dimension and the comprehensive index of the residential land market in the Beijing-ring cities such as Langfang, Zhangjiakou, and Baoding were at the highest level for Hebei Province, while the index of each dimension and the comprehensive index of the residential land markets of the marginal cities such as Qinhuangdao, Chengde, Handan, and Xingtai were at the lower level for Hebei Province and exhibited decreasing trends. These cities may become constraining factors in the coordinated development of the Beijing-Tianjin-Hebei area.

(2) At the county scale: ① The proportion of counties with a healthy residential land market status was greater than 64%, and the residential land market in the entire region was operating well. In 10.92% of the counties, the coordination between the land prices and the national economy improved after 2016, and the rapid increase in residential land prices was an important factor in enhancing the residential land market. ② Local clustering of both the hot and cold state counties occurred. The hot-leaning counties were mainly clustered in the areas around Beijing and Tianjin, and this phenomenon became increasingly obvious after 2016. There was an obvious polarization effect on the residential land market development in the areas around Beijing and Tianjin. To a small extent, most of the cold-leaning counties were clustered in the southern and northeastern fringe areas of the region, and this phenomenon remained obvious after 2016. There is a greater risk of marginalization in the residential land market development in the fringe counties. ③ Regarding the four dimensions or the overall market operation state, both the hot and cold counties exhibited spatial agglomeration characteristics, and most of the county agglomeration was not consistent with the municipal administrative boundaries. The residential land market states of neighboring counties were closely related to each other. The government should fully implement city-based policies and adhere to the regulation and control policies of stable land prices, stable house prices, and stable expectations to promote the healthy development of the residential land market and real estate market. In addition, by decentralizing Beijing's non-capital core functions and supporting the construction of sub-centers in key counties in the peripheral areas, the government can realize the linkage and integration of the regional land market from local areas to the whole region to promote integrated development of the residential land market in the Beijing-Tianjin-Hebei urban agglomeration.

## Acknowledgments

We greatly appreciated Prof. Zhongjiang Feng, Lecturer Man Zhang, and A.P. Yanqing Liang for their valuable comments at various stages of this study. We thank LetPub (www.letpub. com) for its linguistic assistance during the preparation of this manuscript.

## Author Contributions

**Conceptualization:** Xin Chen.

**Data curation:** Xin Chen.

**Funding acquisition:** Can Li, Jingfeng Ge.

**Methodology:** Xin Chen.

**Project administration:** Can Li, Huixia Li.

**Supervision:** Can Li, Huixia Li.

**Writing – original draft:** Xin Chen.

**Writing – review & editing:** Huixia Li, Jingfeng Ge, Wengang Wang, Pengfei An.

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
