## [Decision Letter · Decision Letter 0]

8 Nov 2022

PONE-D-22-29277Evaluation of the operation status and characteristics of the pattern of the residential land market in the Beijing-Tianjin-Hebei region in ChinaPLOS ONE

Dear Dr. Li,

Thank you for submitting your manuscript to PLOS ONE. After careful consideration, we feel that it has merit but does not fully meet PLOS ONE’s publication criteria as it currently stands. Therefore, we invite you to submit a revised version of the manuscript that addresses the points raised during the review process.

We look forward to receiving your revised manuscript.

Kind regards,

Jun Yang

Academic Editor

PLOS ONE

Journal Requirements:

" National Natural Science Foundation of China (project No.41471090) and Science Foundation of Hebei Normal University (project No. L2021B29).Jingfeng Ge is the owner of National Natural Science Foundation of China (Fund No.41471090) and Can Li is the owner of Science Foundation of Hebei Normal University Foundation (project No.L2021B29)."

3. We note that Figures 1, 4a-4h and 5a-b in your submission contain map images which may be copyrighted. All PLOS content is published under the Creative Commons Attribution License (CC BY 4.0), which means that the manuscript, images, and Supporting Information files will be freely available online, and any third party is permitted to access, download, copy, distribute, and use these materials in any way, even commercially, with proper attribution. For these reasons, we cannot publish previously copyrighted maps or satellite images created using proprietary data, such as Google software (Google Maps, Street View, and Earth). For more information, see our copyright guidelines: http://journals.plos.org/plosone/s/licenses-and-copyright.

a. You may seek permission from the original copyright holder of Figures 1, 4a-4h and 5a-b  to publish the content specifically under the CC BY 4.0 license.  

Reviewers' comments:

Reviewer's Responses to Questions

**Comments to the Author**

1. Is the manuscript technically sound, and do the data support the conclusions?

Reviewer #1: Yes

Reviewer #2: Yes

2. Has the statistical analysis been performed appropriately and rigorously? 

Reviewer #1: Yes

Reviewer #2: Yes

3. Have the authors made all data underlying the findings in their manuscript fully available?

Reviewer #1: Yes

Reviewer #2: Yes

4. Is the manuscript presented in an intelligible fashion and written in standard English?

Reviewer #1: No

Reviewer #2: Yes

5. Review Comments to the Author

Reviewer #1: The authors use land market network data and statistical yearbook to evaluate the operation status and characteristics of the pattern of the residential land market in Beijing-Tianjin-Heibei region.Which can promote the coordinated socio-economic development. However, there have some problems need to be revised.

1.In title, the authors should clarify the pattern clearly. Is it spatial pattern, market pattern or others.

2.The content of background is too much.It should be simplified.

3.The content of references are listed simply in literature review. It is lack of summarizing.

4.The authors' name of references should not shown in text.

5.In study area, the logic is too mess, I suggest the authors remove the redundant content.

6.In methods, what is the foundation of classify the market operation into five states?

7.The serial number of sub-title should be supplied. In results, the sub-title is too simple. The authors should revise to high quality sub-title. Such as spatial-temporal evolution of residential land market and so on.

8.The abbreviation of urban residents' disposable income level is UPDI, why the abbreviation of land price-retail price index is not LPRPI?

9.There have two research scale city and county. The authors should illustrate it clearly.

10.How to divide indices of residential land market into five states by ArcGIS. Is it natural breaks(Jenks)or others? The number range of each states also need to supplied. The version of ArcGIS should be illustrated too.

11.The English is really poor. The authors should find a professional instituition to re-edit it. The conciseness of presentation is not enough in this manuscript.

12.In discussion, the content of this section is recommand policy or influencing factor? What are you want to present? The logic is really mess.

13.Some relevant references should be cited as follow.

Response characteristics and influencing factors of carbon emissions and land surface temperature in Guangdong Province,China. Urban Climate,2022:101330.doi:https://doi.org/10.1016/j.uclim.2022.101330.

Seasonal Differences in Land Surface Temperature under Different Land Use/Land Cover Types from the Perspective of Different Climate Zones.Land,2022,11,1122.doi:https://doi.org/10.3390/land11081122.

Spatial and temporal heterogeneity of urban land area and PM2.5 concentration in China. Urban Climate,2022,45:101268. doi:https://doi.org/10.1016/j.uclim.2022.101268.

Spatio-temporal evolution and factors of climate comfort for urban human settlements in the Guangdong-Hong Kong-Macau Greater Bay Area.Front.Environ.Sci.2022,10:1001064. doi:10.3389/fenvs.2022.1001064.

Reviewer #2: This paper presents an interesting topic. The following issues still need to be further improved and explained:

1.The introduction should introduce a series of policies were issued to support the development of residential land market in the Beijing-Tianjin-Hebei region in China, and the introduction should introduce the relevant policy background of implementing the Development policy in this period. What is the meaning of the policy.

2.This is an interesting study based on extensive Statistical material. The authors should focus on improving the readability of the paper, in particular by avoiding data for data's sake.

3. The diagrams of the manuscript need refinement. For example, the map of the study area needs to be labelled with the South China Sea and the Diaoyu Islands; the pictures are blurred and the names of the cities are not visible

4. Analysis of the manuscript's thematic concerns about its characteristics of the pattern, it is recommended to add some references, for example, “Spatial-Temporal Patterns of Network Structure of Human Settlements Competitiveness in Resource-Based Urban Agglomerations”，“Spatial Responses of Ecosystem Service Value during the Development of Urban Agglomerations ” and “Morphological and functional polycentric structure assessment of megacity: An integrated approach with spatial distribution and interaction”.

6. PLOS authors have the option to publish the peer review history of their article (what does this mean?). If published, this will include your full peer review and any attached files.

Reviewer #1: No

Reviewer #2: No

---

## [Author Response · Author response to Decision Letter 0]

19 May 2023

For the additional requirements of editing, our answers are as follows:

1. We checked the manuscript to make it conform to the style requirements of PLOS ONE.

2. The funders played a relevant role in this article, Jingfeng Ge has done Writing-review & editing; Can Li has done project management and supervision.

3. The figures in this paper is made according to the picture released by the Ministry of Natural Resources of China. The review number is GS (2016) No. 1610, which does not involve copyright protection.

In addition, for the language questions raised by reviewer, we put the proofs of the article retouching in the ‘other’ folder.

Reviewer #1

Comments:

1.In title, the authors should clarify the pattern clearly. Is it spatial pattern, market pattern or others.

Reply: The authors appreciated the reviewer’s kindly and careful comments and suggestions. What’s more, the authors added 'space-time pattern' to the title of this article and changed it to '' ‘Evaluation of the operation status of the residential land market in Beijing-Tianjin-Hebei region of China and Its spatiotemporal pattern characteristics’ Thanks.

2.The content of background is too much. It should be simplified. 

Reply: Thanks for your constructive suggestion. The authors have simplified the introduction.

3.The content of references are listed simply in literature review. It is lack of summarizing.

Reply: Thanks for your insightful and interesting question. We have summarized and classified the references, and added an inductive evaluation at the end of the paragraph.

4.The authors' name of references should not show in text.

Reply: Thank you for your correction. We have deleted the author's name from the references.

5.In study area, the logic is too mess, I suggest the authors remove the redundant content.

Reply: Thank you for your valuable suggestions. We have deleted irrelevant content, such as land price and economic development level. The changed contents are displayed in Lines 136-145 in the Revised Main Manuscript.

6.In methods, what is the foundation of classify the market operation into five states?

Reply: We made a detailed description in the second part of the method, that is, using K-S test and 3σ methods for defining five states of " overheated, slightly hot, healthy, slightly cold, overcooled " were formulated. The results of K-S test are shown in Table 2, and Table 3 proves the application of 3σ methods. Finally, the interval division of five states is shown in Table 4. Thanks.

7.The serial number of sub-title should be supplied. In results, the sub-title is too simple. The authors should revise to high quality sub-title. Such as spatial-temporal evolution of residential land market and so on.

Reply: Thank you for your suggestion, and we set the short title as ‘Evaluation on the operation of residential land market’. And prepare to modify in the submission system.

8.The abbreviation of urban residents' disposable income level is UPDI, why the abbreviation of land price-retail price index is not LPRPI?

Reply: Thanks for your insightful and interesting question. We abbreviated 'land price detail price index' as ‘LPRPI’, and revised lines 201, 271 and 528 in the revised manuscript.

9.There have two research scale city and county. The authors should illustrate it clearly.

Reply: Thank you for your valuable comments. It is necessary to explain the two research scales. In the abstract part of this paper, it is explained that the data collection and calculation are carried out by taking the county as the unit. At the beginning of the results section, it is stated that the average value of the market state index of residential land in all counties within the jurisdiction of each city is calculated as the market state index of urban residential land, taking cities as the unit. Because taking cities as a unit has a more concise role in explaining the differences between regions.

10.How to divide indices of residential land market into five states by ArcGIS. Is it natural breaks (Jenks)or others? The number range of each states also need to supplied. The version of ArcGIS should be illustrated too.

Reply: This is an important question. We do not use natural breaks (Jenks). In ArcGIS, we use specific value intervals to divide five states. The specific value intervals can be seen in Table 4. In addition, we have updated the pictures. According to your requirements, we have placed the range of five levels in the legend area of Fig 4 and Fig 5. Finally, the version number of ArcGIS 10.2 is added in line 445 and line 504 of the revised manuscript.

11.The English is really poor. The authors should find a professional instituition to re-edit it. The conciseness of presentation is not enough in this manuscript.

Reply: Thank you for your comments. We polished the article and put the polish proof in other folders.

12.In discussion, the content of this section is recommand policy or influencing factor? What are you want to present? The logic is really mess.

Reply: Thanks a lot for your careful comments. It may be that the subtitle 'Comparison of Residential Land Market Status and Policy Orientation in Beijing, Tianjin and Hebei' cannot clearly explain the content of the discussion. We have revised two subtitles and changed 'policy orientation' to 'policy recommendations'. They are placed in lines 474-475 and 517-518 of the revised manuscript. It is worth mentioning that our discussion has two parts, representing two dimensions, namely, the dimensions of the three provincial regions in Beijing, Tianjin and Hebei and the dimensions of the county level. Each part also discusses the current situation and policy recommendations. According to the suggestion, we adjusted some statements. 

13.Some relevant references should be cited as follow.

Reply: Thank you for your suggestion. After referring to the corresponding literature, we added the literature to our article.

Reviewer #2:

1.The introduction should introduce a series of policies were issued to support the development of residential land market in the Beijing-Tianjin-Hebei region in China, and the introduction should introduce the relevant policy background of implementing the Development policy in this period. What is the meaning of the policy.

Reply: The authors appreciated the reviewer’s precise and insightful comments and suggestions, which could significantly improve the logicality, fluency and accuracy of the manuscript. The introduction of the policy part is to highlight the importance and necessity of the current research in Beijing Tianjin Hebei region. At the same time, we have deleted the redundant parts. Make the logic clearer. Thanks.

2.This is an interesting study based on extensive Statistical material. The authors should focus on improving the readability of the paper, in particular by avoiding data for data's sake.

Reply: Thank you for your constructive suggestions. We have the same feeling about the data display as you do, and we do not want to display too much data. We have filtered the data display as much as possible, and explained the data and described the readability in text after the data. We really hope to get your understanding.

3. The diagrams of the manuscript need refinement. For example, the map of the study area needs to be labelled with the South China Sea and the Diaoyu Islands; the pictures are blurred and the names of the cities are not visible.

Reply: Thanks for your careful suggestions. We have now updated the map of the study area and adjusted other pictures appropriately.

4. Analysis of the manuscript's thematic concerns about its characteristics of the pattern, it is recommended to add some references, for example, “Spatial-Temporal Patterns of Network Structure of Human Settlements Competitiveness in Resource-Based Urban Agglomerations”，“Spatial Responses of Ecosystem Service Value during the Development of Urban Agglomerations ” and “Morphological and functional polycentric structure assessment of megacity: An integrated approach with spatial distribution and interaction”.

Reply: Thank you for your suggestions on the references. We have added some of the l references you suggested. If there are still shortcomings, you are welcome to point them out clearly.

---

## [Decision Letter · Decision Letter 1]

4 Sep 2023

Evaluation of the operation status of the residential land market in Beijing-Tianjin-Hebei region of China and Its spatiotemporal pattern characteristics

PONE-D-22-29277R1

Dear Dr. Li,

We’re pleased to inform you that your manuscript has been judged scientifically suitable for publication and will be formally accepted for publication once it meets all outstanding technical requirements.

Kind regards,

Jun Yang

Academic Editor

PLOS ONE

Additional Editor Comments (optional):

Accept

Reviewers' comments:

Reviewer's Responses to Questions

**Comments to the Author**

1. If the authors have adequately addressed your comments raised in a previous round of review and you feel that this manuscript is now acceptable for publication, you may indicate that here to bypass the “Comments to the Author” section, enter your conflict of interest statement in the “Confidential to Editor” section, and submit your "Accept" recommendation.

Reviewer #1: All comments have been addressed

2. Is the manuscript technically sound, and do the data support the conclusions?

Reviewer #1: Yes

3. Has the statistical analysis been performed appropriately and rigorously? 

Reviewer #1: Yes

4. Have the authors made all data underlying the findings in their manuscript fully available?

Reviewer #1: Yes

5. Is the manuscript presented in an intelligible fashion and written in standard English?

Reviewer #1: Yes

6. Review Comments to the Author

Reviewer #1: All the problems have been addressed. The conclusions are too long, it should be refined and shorted further.

7. PLOS authors have the option to publish the peer review history of their article (what does this mean?). If published, this will include your full peer review and any attached files.

Reviewer #1: No

---

## [Editor Report · Acceptance letter]

13 Sep 2023

PONE-D-22-29277R1 

Evaluation of the operation status of the residential land market in Beijing-Tianjin-Hebei region of China and Its spatiotemporal pattern characteristics 

Dear Dr. Li:

I'm pleased to inform you that your manuscript has been deemed suitable for publication in PLOS ONE. Congratulations! Your manuscript is now with our production department. 

Kind regards, 

on behalf of

Dr. Jun Yang 

Academic Editor

PLOS ONE